Genetic diversity, functional properties and expression analysis of NnSBE genes involved in starch synthesis of lotus (Nelumbo nucifera Gaertn.)

Zhu Fenglin 1 2 3
Sun Han 1 2 3
Diao Ying 1 3
Zheng Xingwen 1 2 3 4
Xie Keqiang 4
Hu Zhongli huzhongli@whu.edu.cn 1 2 3
1 College of Life Sciences, Wuhan University , Wuhan , China
2 State Key Laboratory of Hybrid Rice , Wuhan , China
3 Hubei Lotus Engineering Center , Wuhan , China
4 Guangchang Bailian Institute of Jiangxi Province , Guangchang , China
Eslami Gilda
Electronic publication date: 2019 Sep 25
Publication date: 2019
Volume: 7
Electronic Location ID: e7750
Received 2019 Apr 18; Accepted 2019 Aug 25
Copyright: ©2019 Zhu et al.
Copyright year: 2019
Copyright holder: Zhu et al.
License: This is an open access article distributed under the terms of the Creative Commons Attribution License, which permits unrestricted use, distribution, reproduction and adaptation in any medium and for any purpose provided that it is properly attributed. For attribution, the original author(s), title, publication source (PeerJ) and either DOI or URL of the article must be cited.
License URL: https://creativecommons.org/licenses/by/4.0/

Keywords: Amylopectin, Affinity activity, Lotus, NnSBE, Relative expression, Homozygous haplotype

Funding: National Science and Technology Supporting Program 2012BAD27B01 Technology Innovation Project of Hubei Province of China 2019ABA108 This work was supported by the National Science and Technology Supporting Program (No. 2012BAD27B01) and the Technology Innovation Project of Hubei Province of China (No. 2019ABA108). The funders had no role in study design, data collection and analysis, decision to publish, or preparation of the manuscript.

==============================
Background

Starch branching enzyme (SBE) is one of the key enzymes in starch biosynthetic metabolism, determining amylopectin structure.

Methods

Full length coding sequences (CDS) of SBE genes were cloned using reverse transcription PCR (RT-PCR) technology, and neighbor-joining (NJ) tree was used for phylogenetic analysis. Single nucleotide polymorphisms (SNPs) were determined to assess the genetic polymorphisms and variation indexes between individuals and clusters. Quantitative real time PCR (qRT-PCR) was performed to analyze the spatial and temporal expression of NnSBE genes. The effect of NnSBE genes on amylopectin’s fine structures was explored using affinity and the enzyme activity analysis of two isoforms in amylopectin and amylose.

Results

In this study, two SBE family genes, NnSBEI and NnSBEIII, were identified in lotus (Nelumbo nucifera Gaertn.). Phylogenetic analysis sorted NnSBEI into SBE family B and NnSBEIII into SBE family A. UPGMA phylogenetic tree divided 45 individuals of lotus into three classes. The homozygous haplotype (A G G A G) of NnSBEIII was observed in seed lotus. During the seed embryo development stage, NnSBEIII reached the peak in the middle of the development stage, while NnSBEI increased in the mid-late developmental stage. The different affinity activity of the two isozymes binding amylopectin and amylose assay indicated NnSBEI has higher activity and wider affinity.

Discussion

Genetic diversity showed that NnSBE genes received artificial selection during the process of cultivation and domestication in lotus seeds. Furthermore, the expression pattern and affinity activity analysis indicated that NnSBE genes were related to the chain length of amylopectin.

Introduction

Lotus (Nelumbo nucifera Gaertn) is an ancient perennial aquatic plant and important crop in Asia. Archaeological research has estimated that the history of lotus is more than 7,000 years old, and it has been cultivated for more than 2000 years (Guo, 2009; Shen-Miller, 2002). In general, lotus are grouped into three clusters corresponding with the three different important organs, namely ornament lotus, seed lotus and rhizome lotus, respectively (Guo, 2009; Wu et al., 2007). As a type of aquatic plant with high photosynthetic efficiency and high carbon conversion, lotus has high starch content. Because of their high amount of starch, the edible rhizomes and seeds play a key role in a daily diet and cultural activities (Chen et al., 2007). Studies have shown that fresh rhizomes, on average, are comprised of 10–20% starch in their total fresh weight, while the amount is 40–60% in mature seeds (Shen-Miller et al., 1995).

Starch is an important polysaccharide and the major form of carbohydrate storage in plants (Slattery, Kavakli & Okita, 2000). It is a necessary part of the human diet in terms of nutrition and calories. Starch is comprised of two glucan polymers, amylose and amylopectin, which have different characteristics because of their starch molecular structure. Amylose and amylopectin synthesis are regulated by the coordinated action of a series of enzymes. AGPase produces substrate that plays a role in the synthesis of amylose and amylopectin. Amylose synthesis is mainly controlled by granule-bound starch synthase (GBSS), while amylopectin is generated by the successive work of starch synthase (SS), starch branching enzyme (SBE), and debranching enzyme (DBE) (Fujita et al., 2008; Jeon et al., 2010; Subasinghe et al., 2014; Tetlow Ian & Emes Michael, 2014).

Starch-branching enzymes, formerly known as Q-enzymes, have dual catalytic functions and determine the structure of amylopectin. SBE cleaves the internal a-1,4 linkage from polyglucans and then transfers the reducing ends to C-6 hydroxyls to generate a-1,6-branch linkage in the liner chain (Martin & Smith, 1995). Their catalytic function not only catalyzes the formation of a new branch, but they also add new nonreducing ends in the starch molecule. Thus, SBE determines the branching pattern in amylopectin, which is believed to affect the fine structure of plants and influence the amount of starch (Satoh et al., 2003).

SBEs have been researched in various plants. In particular, two cDNA coding PsSBEs were identified from the embryo of Pisum sativum by Burton Rachel et al. (1995). Later, two or three SBE members were identified in various plants and were classified into SBE A and SBE B based on phylogenetic analysis. SBE A and SBE B play different roles in the influence of amylopectin synthesis. In the process of starch synthesis, the SBE A family tends to amylopectin, while SBE B family show higher affinity for amylose. SBE A prefers amylose as a substrate and predominantly transfers relatively longer chains (>14DP), while SBE B tends to amylopectin and transfers shorter chains (<14DP) (Guan et al., 1997). In addition, different gene expression patterns of SBE A and SBE B were detected with a range of species, indicating that SBE B is expressed earlier than SBE A in the development stage (Gao et al., 1996; Larsson et al., 1998; Morell et al., 1997; Mutisya et al., 2003).

To fit the rapid development of the lotus processing industry, the work of breeding lotus with high starch content is extremely urgent. Therefore, many studies have focused on the starch of lotus. The ADPase and GBSS genes of lotus were isolated and characterized, but little information is currently available about the NnSBE gene (Cheng et al., 2014; Liang-Jun et al., 2006; Lu et al., 2012). For researching the NnSBEs further, this study undertook the isolation of cDNA and genomic clones by encoding two related SBE genes from lotus. Then, genomic variation and evolution were analyzed using DNA and protein sequences of the two NnSBEs. Expression patterns and the affinity of two isozymes were described, as well as the enzyme activity of both NnSBEs in development and different tissues. These data provide valuable information for understanding the processes involved in starch synthesis, and offer some fundamental information for further study about improving the edible quality of lotus.

Material and Methods

Plant material and treatments

Nelumbo nucifera cv. Taikong lotus 36, the highest strain of selective lotus breed after space mutagenesis, was selected in this study (Wu et al., 2007). Seeds were sprouted by soaking in water for germination. Five days after, plants were provided with 50 cm depth pots in a greenhouse for the entire growing season. Functional leaves, petiole, rhizomes and roots were separately collected in the 8th week, 10th week, 12th week. Seeds from plants at 12, 16, 20, 24 and 28 DAF (days after fertilization) were also collected from the Taikong lotus 36 in the genetic experimental base of Wuhan University. Various materials from lotus were quick-frozen in liquid nitrogen, then stored at −80 °C for next manipulation. The fresh leaves of 45 lotus individuals (Supplement 1) were collected from the genetic experimental base of Wuhan University and stored in silica gel (Sinopharm, China).

RNA isolation and cDNA synthesis

Total RNA and genomic DNA were isolated from plant tissue samples using a plant RNA extraction kit and Plant Genomic DNA Kit (TIANGEN, Beijing, China), according to their respective operation manuals. The quantities were subjected to 1% agarose gel electrophoresis and the results were analyzed using a UV Transilluminator (Eppendorf, Hamburg, Germany). The first-strand cDNA of all materials was constructed using the FastKing RT Kit (with gDNase) (TIANGEN). The products were stored at −20 °C for later use.

Characterization of a genetic polymorphism

Genetic polymorphism of NnSBE cDNA from 45 individuals of lotus were also analyzed using the gene-specific primers. PCR reactions were conducted in 50 µl volumes containing two µl of total cDNA, five µl of 10 ×PCR buffer, five µl of two mM dNTPs mixture, three µl of 25 mM MgSO4, 1.6 µl of 10 pM of each primer, one µl of one U/µl KOD DNA polymerase (TOYOBO, Osaka, Japan) and 30.8 µl ddH2O, for a total volume of 50 µl. Amplification conditions followed the two-step amplification procedure: 94 °C for 2 min, 36 cycles of 98 °C for 10 s, (Tm)  °C for 30 s, and 68 °C for 1 min. The sizes of the PCR products were assessed using 1.0% agarose gel electrophoresis. PCR products were sequenced by Sanger sequencing (Augct, Beijing, China). Sequences were aligned by DNAman to find molecular markers.

Bioinformatics analysis

All sequence data were obtained from the National Center for Biotechnology Information (NCBI) GenBank database (http://www.ncbi.nlm.nih.gov/). All primers were designed by Primer Premier 5.0v. Genomic structures were performed with the program Gene Structure Display Server 2.0 (http://gsds.cbi.pku.edu.cn/) (Hu et al., 2015). The GO (Gene Ontology, http://www.geneontology.org) was used for annotation of gene products. The conserved domains were predicted by Pfam server (http://pfam.xfam.org/search/sequence;) (Finn et al., 2016). All sequences were aligned using Cluxal-X(Thompson et al., 1997) and DNAman. Subsequently the phylogenetic neighbor-joining tree was generated by MEGA6.0v (Tamura et al., 2013). The bootstrap consensus tree inferred from 1,000 replicates was taken to depict the evolutionary history of the analyzed taxa. The HO, HE, Shannon Index was calculated by popGen32 (Nei, 1978; Nei & Li, 1973; Nei & Li, 1979). A dendrogram of the cluster analysis was based on Nei’s genetic distance using the UPGMA method.

Real-time PCR analysis

The cDNA of reverse transcription was diluted for 1:10 and used for RT-PCR and qRT-PCR assays. A house-keeping gene, CYP (cyclophilin, GenBank accession no. EU131153), was selected as the reference gene in this experiment. The primers of CYP were based on the sequence of CDS and designed by Primer Primer 5.0 (Table 1). The total PCR reaction mixture contained 2 µl cDNA (1:10 diluted), 0.4 µl forward primer (10 M), 0.4 µl reverse primer (10 M), 10 µl 2 ×SYBR qPCR Master Mix (with ROX Premixed) (Vazyme, China) and 7.2 µl ddH2O, for a total volume of 20 µl. For the qRT-PCR experiment, the two-step amplification procedure was used: 95 °C for 10 min, followed by 40 cycles of 95 °C for 15 s and 60 °C for 1 min. The relative gene expression data was calculated using the 2−ΔΔCt method with the guidance of the StepOne software v2.1 (ABI, US). The experimental design followed MIQE (minimum information for publication of quantitative real-time PCR experiments) (Bustin et al., 2010; Bustin et al., 2009). All measurements were processed three times for biological and technical paralleled repetition.

Table 1 Primers used in qRT-PCR.

Gene	Sequence forward & reverse primers (5′-3′)	Amplicon length (bp)	R2	Primer efficiency	
qCYP-F	GTACCCAGAAGAATGCCCTA	102	0.998	96.222	
qCYP-R	ATGAAGCCCTTGATGACTCG				
qNnSBEI-F	GTAGACCATTTCACATCGC	114	0.999	93.132	
qNnSBEI-R	TAATAAGCCACACATGTACGAG				
qNnSBEIII-F	TATGCATGGCTAGTTCCAC	110	0.999	86.246	
qNnSBEIII-R	TTATGCCAAAATGCCTCGT				

Construction of plasmids

The full length CDS of NnSBEI and NnSBEIII, which were cloned from lotus, were subcloned into the plasmids pET-28a and pET-32a respectively. The pET-28a-NnSBEI was generated using the following primers: 5′-CCCAAGCTTATGTACAGTTTTTCTGGGT-3′(HindIII site underlined), and 5′-CATGCCATGG TCAGTCATCCAATCCCA-3′(Nco I site underlined). The pET-32a-NnSBEIII was obtained by primers: 5′-CGGGATCCATGG CTACTACAGTTGCGCT-3′(BamHI site underlined) and 5′-GCGATATCTCATATTCG CAAAATCCGAG-3′(EcoR V site underlined). The full length and sequence of the nucleotide acid of the inserted gene in the extracted recombinant plasmid were completely identical to the genes. Then, the different recombinant plasmids were transformed into Escherichia coli BL21 (DE3) (TransGen Biotech, Beijing, China) for expressing and into DH5α for storing.

Enzyme activity assay

Cheng et al.’s protocol of measuring SBE activity was used in this study (Cheng et al., 2001). First, total enzyme isolation: about 0.4 g powder of plant material was added into a 4 ml HEPES–NaOH (PH7.5) buffer. After proper shaking, the tube was centrifuged at 10,000 g for 15 min, and supernatant was extracted to check the enzyme activity of SBE. We took 100 µl dilution, added 1,280 µl of HEPES–NaOH (PH7.5) buffer, and 120 µl 0.75% of soluble starch. Then we incubated the total reaction mixture at 37 °C, and gave it a water bath for 20 min. Next, we gave it a boiling water bath for 1 min to stop the reaction, diluted the mixture with two ml water (containing 0.2% HCl), added 150 µl of an iodine solution (0.1%I2-1%KI), blending it at room temperature for 10 min. The boiled crude enzyme was used as the control. Finally, we measured absorbance at 660 nm. The enzyme activity of SBE was expressed as a percentage of decrease in absorbance at the wavelength of 660 nm to be compared with the control. All measurements were processed three times for biological and technical paralleled repetition.

The affinity of two isozymes that bind amylopectin and amylose assay

Recombinant proteins were expressed in the Escherichia coli BL21(DE3) cells. The transformed cells were grown in LB at 37 °C until the OD600 reached 0.6–1.0, using IPTG to induce expression with a final concentration of 0.5 mM, and were further incubated at 16 °C for 12 h. Cells were harvested by centrifugation at 12,000 rpm for 5 min at 4 °C. The products were suspended in a buffer containing 10 mM PBS in the radio of 200:1 (ml/g), and cells were disrupted (Diagenode, Liege, Belgium) by sonication for 25 min and centrifuged for 5 min at 4 °C. We took 30 µl supernatant, added 0, 12.5, 16, 20, 25, 35 µll 0.75% amylopectin (Sigma-Aldrich, St. Louis, MO, USA) or amylose (Sigma), and added HEPES–NaOH (PH7.5) buffer into 90 ml. Then the total reaction mixtures were incubated at 37 °C for 30 min. We diluted the mixture with 100 µll water (containing 0.2% HCl), and added 10 µll iodine solution (0.1%I2-1%KI), blending at room temperature for 10 min. Finally, we measured absorbance of amylopectin at 560 nm and amylose at 620 nm. The blank pET-28a and pET-32a which were transformed into BL21 and cultured in the same condition were used as the control. The affinity of the two isozymes that bind amylopectin and amylose were measured by enzyme activity. All measurements were processed three times for technical repetition and 2∼3 paralleled repetition.

Results

Cloning of NnSBE genes

Based on the reference sequence of whole-genome sequencing, the CDS of two SBE genes were obtained using RT-PCR technique from the embryo of the lotus seed. The CDS of NnSBEI (FJ592190.1) was 2,577 bp, and the CDS of NnSBEIII (XM_010254179) was 2691bp (Fig. 1). Complete genomic structures of NnSBEI (comprising of 14 exons and 13 introns) and NnSBEIII (comprising of 21 exons and 20 introns) were separately distributed over 14.3 and 32.1 kb. The gene structures are shown in Fig. 2 and the gene information is shown in Table 2. The cDNA of NnSBEI showed the highest (85%) identity with Castanea mollissima, and 80–84% identify with SBEI genes of other plant species, while NnSBEIII showed the highest (85%) identity with Juglans regia and 80–84% identity with the others.

Figure 1 The amplication of cDNA fragments of NnSBEs.

The amplication of cDNA fragments of NnSBEs NnSBEI and NnSBEIII using RT-PCR experiments. The products were analyzed by electrophoresis on 1.0% agarose gels, DL 5000 DNA marker.

Figure 2 The gene structure of NnSBEI and NnSBEIII.

The blue boxes represent UTR, black boxes represent the exons, thick line represent intros.

Table 2 Gene information of NnSBEI and NnSBEIII.

Gene name	Gene ID	Genomic length(kb)	ORF length (bp)	Number of exons	Number of intros	Number of amino acid (Aa)	
NnSBEI	104,603,742	14.3	2,352	14	13	783	
NnSBEIII	104,594,060	32.1	2,691	21	20	896	

The deduced protein of NnSBEI comprised 858 amino acids with a predicted molecular mass of 97.144 kDa, and NnSBEIII comprised 896 amino acids with a predicted molecular mass of 103.135 kDa. GO analysis indicated NnSBE participated in biological processes (GO:0005978) and one molecular function (GO:0043169; GO:0003844; GO:0004553). Pfam analysis showed that NnSBEs have three domains of secondary structure: the central catalytic A domain, an N-terminal domain and the C-terminal domain. The central catalytic A-domain of SBEs is α-amylase which covered four conserved amino acid regions and six active sites spread in those conserved regions respectively. The characterization of catalytic A-domain showed significant homology (47.83%) between NnSBEI and NnSBEIII, but highly dissimilar sequences in N-terminus (21.84%) and C-terminus (26.47%).

Phylogenetic analysis

SBE protein sequences from 20 other species were used for phylogenetic analysis, shown in Fig. 3. The NJ phylogenetic tree indicates that SBEs in these plants could be classified into two families: SBE A and SBE B. SBE A was further divided into four classes: algae SBEII, dicot SBEII, monocot SBEII and SBEIII, and SBE B was subdivided into algae, dicot SBEI, monocot SBEI. NnSBEI belonged to the dicot SBEI of family B; NnSBEIII belonged to the SBEIII class of family A. Evolutionary divergence reveal that SBEIII came before SBEI, and differentiation of NnSBE occurred before that of the other dicots, but later than the monocots.

Figure 3 Phylogenetic analysis of the SBEs family.

The dendrogram was constructed using MEGA6.0 software with the neighbor-joining method. Sequences aligned included the SBE protein sequences of 20 other species, which identified or predicted from NCBI database were used for phylogenetic analysis. Different color regions represent two subclades of SBEs: the blue region cover SBE A family, and the gray cover SBE B family. The NnSBEs were labeled with “●”.

Characterization of genetic diversity

Genetic diversity was identified based on the ORF of two NnSBE genes from 45 lotus individuals belonging to four clusters (Supplement 1). This revealed six polymorphic sites from the coding region: five SNPs in NnSBEIII and one in NnSBEI. The SNPs of NnSBEIII was analyzed and the detailed parameters of genetic diversity are listed in Table 3. Three SNPs resulted in missense mutations which changed the amino acid sequence, while the other two SNPs were synonymously mutated. The observed heterozygosity ranged from 0.2 to 0.4444, and the expected heterozygosity ranged from 0.4012 to 0.4583. The Shannon-Wiener Index ranged from 0.5908 to 0.6508. In further association analysis of starch content (amylopectin, amylose, total starch) in lotus seeds, no SNP showed highly significant associations. Total of 12 genotypes were detected in those 45 individuals, each cluster had different several kinds of genotypes (Supplemental Information 1). A homozygous genotype (AA GG GG AA GG) of NnSBEIII was observed in most individuals in seed lotusand a haplotype (A G G A G) was identified from seed lotus. Cluster analysis according to the UPGMA method is shown in Fig. 4. The 45 individuals of lotus were copolymerized into three classes. The first class can be divided into two subclasses with a similarity coefficient of 9.8, and most seed lotus showed together in a same subgroup. These results paved a way to apply the useful allelic variations or gene haplotypes in cultivar lotus and quality breed programs.

Table 3 Genetic diversity based on SNPs of NnSBEIII in the test population.

SNPs of NnSBEIII	Aa	Allelic frequency	Genotypic frequency	HO	HE	Shannon index	
c.282 G>A	M	A	0.6778	AG	0.2444	0.2444	0.4368	0.6285	
		G	0.3222	AA	0.5556				
				GG	0.2				
c.535 G>A	E/K	A	0.3111	AG	0.3556	0.3556	0.4286	0.6200	
		G	0.6889	AA	0.1333				
				GG	0.5111				
c.880 G>A	V/L	A	0.3556	AG	0.4444	0.4444	0.4583	0.6508	
		G	0.6444	AA	0.1333				
				GG	0.4222				
c.2308 G>A	O/N	A	0.7000	AG	0.2889	0.2889	0.4200	0.6109	
		G	0.3000	AA	0.5556				
				GG	0.1555				
c.2559 G>A	G	A	0.2778	AG	0.2000	0.2000	0.4012	0.5908	
		G	0.7222	AA	0.1778				
				GG	0.6222				
Notes.

HO the observed heterozygosity

HE the expected heterozygosity

Expression pattern

The temporal and spatial expression of both genes were analyzed to investigate their expression patterns. The results of qRT-PCR demonstrated that the transcriptional expression level of NnSBEI and NnSBEIII was detectable in all tissues. Tissue-specific expression analysis showed that the highest transcript levels of NnSBEI and NnSBEIII were observed in leaves. The transcript level of NnSBEI was higher in rhizomes while NnSBEIII expressed strongly in petioles. The expression profile of NnSBEI in the rhizome showed strong temporal differences, and the relative expression level increased gradually from initial development to late stage. NnSBEIII showed temporal differences in the petiole, which enhanced in the middle and decreased in the late stage (Fig. 5). During the seed embryo developing stage, NnSBEI and NnSBEIII expressed significant differences. Transcripts of NnSBEIII reached the peak at 20 DAF in the middle of development stage, and then decreased gradually, while NnSBEI increased in 16 DAF to 28 DAF and expressed strongly in the mid-late developmental stage (Fig. 6). In the peak stage, NnSBEI expressed ten times higher than NnSBEIII.

Dynamic changes of enzyme activities

The dynamic changes of enzyme activity of the SBEs were analyzed in different temperatures, tissues and developing stages. Incubation in different temperatures revealed that the highest enzyme activity was generated under 37 °C. The enzyme activity of inter-organizational measurement and assessment showed the highest catalytic activity in leaves, followed by rhizomes and petioles which increased steadily, and activity in the roots was the weakest. During seed development, enzyme activity increased rapidly, and the activity peak appeared at 20 DAF, then decreased slightly (Fig. 7).

The affinity of two isozymes that bind amylopectin and amylose

To explore the affinity activity, recombinant DNA techniques were used to generate the pET-28a-NnSBEI and pET-32a-NnSBEIII plasmids. Testing of the inducible expression vector showed that both proteins were soluble, and the apparent molecular weights of pET-28a-NnSBEI and pET-32a-NnSBEIII were about 105 kDa and 120 kDa (with His tag) respectively. The affinity activity of NnSBEI and NnSBEIII to amylopectin and amylose were assayed at various starch concentrations. As we can see in Fig. 8, the activity of both isozymes correlated with the types and concentration of substrate. Rate of reaction changed with the increase of the substrate concentration until the enzyme was saturated. NnSBEI showed higher affinity activity when the substrates were amylose and amylopectin. However, another isozyme, NnSBEIII, only worked on amylopectin, and the branching efficiency was only half of NnSBEI.

Figure 4 Dendrogram of 45 individuals lotus based on SNP data of NnSBEIII.

Dendrogram based Nei’s genetic distance and the UPGMA method was using for Cluster analysis. The 45 individuals of lotus were copolymerized into three classes. The number of individuals were shown in the Supplemental Information 1 and SNPs data were shown as colormetric distinction.

Figure 5 Tissue-differential expression of the NnSBEs.

The column chart confirmed expression pattern of leaves, petiole, rhizomes, roots which collected in 8th week after sowing at the early swelling stage (A, D), collected in 10th week after sowing at the middle swelling stage (B, E), collected in 12th week after sowing at the later swelling stage (C, F) in lotus. The gray boxes represent NnSBEI and the lightgray represent NnSBEIII. Error bars indicate standard error (n = 3) (p < 0.05).

Figure 6 Expressed tendency of the NnSBEs during developing seed.

Relative expression of the NnSBEs were analyzed at 12, 16, 20, 24, 28 DAF. The gray boxes represent NnSBEI (A) and the light gray boxes represent NnSBEIII. (B) Error bars indicate standard error (n = 3) (p < 0.05).

Figure 7 Enzyme activity of NnSBE.

Tissue-differential enzyme activity of NnSBE in leaves, petiole, rhizomes, roots which collected in 8th week after sowing at the early swelling stage, 10th week after sowing at the middle swelling stage, 12th week after sowing at the later swelling stage, and the different colors (black, red, blue, purple) represent various organs(leaf, petiole, rhizome and root) in an orderly way from top to bottom (A). Enzyme activity of developing seed of NnSBE at 12, 16, 20, 24, 28 DAF, the green lines represent seed (B). Error bars indicate standard error (n = 3) (p < 0.05).

Figure 8 The rate of change in absorbance about NnSBEI and NnSBEIII reacted among different substrate concentration.

The substrate is amylopectin (A); the substrate is amylose (B). The black lines represent NnSBEI; the red line represents NnSBEIII.

Discussion

Starch is an important edible component in lotus, and the starch branch enzyme is involved in the synthesis of amylopectin. In this study, we focused on the SBE family genes in lotus, and revealed two isoforms of NnSBEs: NnSBEI and NnSBEIII. In this study, SBE phylogenetic tree included the SBE gene families from algae, monocots and dicots according to the protein sequences of the sequenced species. Two or three SBE isoforms were isolated from different plants, and divided into two families: family A and family B (Fig. 4). NnSBEI belonged to family B, which has been reported in a range of species. NnSBEIII belonged to family A, which has only been identified in maize, wheat, rice and lotus, and little research has been done about this isoform (Kang et al., 2013; Tian et al., 2009).

Over the last decades, many investigations have been devoted to exploring genomic variation and evolution among different germplasms. Whole genome re-sequencing reveals the evolutionary patterns of sacred lotus, rhizome lotus had the lowest genomic diversity and a closer relationship to wild lotus, whereas the genomes of seed lotus and ornament lotus were admixed (Huang et al., 2018). In this study, genetic diversity and genetic variation of NnSBEIII were investigated among 45 individuals from four lotus subgroups. Wild lotus had the higher genomic diversity, rhizome lotus was admixed with wild and flower lotus respectively, and lotus seed formed a homozygous single haplotype, which might be the result of continuous selection in the cultivation process of lotus in the long-term evolution process (Doebley, Gaut & Smith, 2006; Yue, Melamud & Moult, 2006). This different performance induces speculation that the NnSBE gene has more effect on starch accumulation in seed lotus, but less in rhizome lotus.

Starch synthesis and accumulation are closely related to photosynthesis. It was found that the AGPase and GBSS genes of lotus, related to starch synthesis, were expressed higher in leaves (Cheng et al., 2014; Lu et al., 2012). As the tissue of photosynthesis, leaves support the first step of synthesizing sugar and meet the carbon demand faster and it is the major tissue for accumulating transient starch. The results of qRT-PCR showed that the two NnSBEs were expressed throughout plant tissues. NnSBEI was strongly activated in the leaves and petiole; NnSBEIII was highly expressed in the leaves and stems (Fig. 4). The transcriptional level kept rising during the swelling stage in the petiole and rhizome, which represent the photosynthetic tissue and storage tissues respectively. It is possible to adapt to the synthesis of transient starch in photosynthetic tissue, and it is consistent with reserve starch synthesis in storage organs during starch rapid development. This spatial and temporal variation from top to bottom may be related to the process and the transport of starch synthesis to achieve higher efficiency of starch accumulation. Such expression patterns of temporal and spatial controls have also been found in many other species (Tetlow, 2010). This spatial and temporal expression pattern of NnSBEs help amylopectin to adapt requirements at different developmental stages.

During evolution, SBE A family is inclined to the amylopectin branch, while SBE B family is more sensitive to amylose. Likewise, this experiment showed that NnSBEI has higher catalytic activity for amylose and amylopectin, while NnSBEIII expressed catalytic activity only when the substrate was amylopectin. Protein sequences of the SBE genes among a range of species revealed that the SBEI and SBEIII subunits showed homology in their α-amylase catalytic domain, but highly dissimilar sequences in N-terminus and C-terminus. Construction of chimeric enzymes out of maize branching enzymes found that the N-terminal determined the specificity of the transferred chain length, and the C-terminal domain participates in the specificity of the substrate (Commuri & Keeling, 2001; Hong, Mikkelsen & Preiss, 2001; Kuriki, Stewart & Preiss, 1997). Therefore the N-terminal region of SBE is essential for maximum enzyme activity and thermostability (Hamada et al., 2007), and C-terminus determines both substrate preference and maximal catalytic activity (Hong & Preiss, 2000). We speculated that the difference between enzymatic characteristics of the two subunits are determined by the N-terminus and C-terminus. These differences met the requirements of amylopectin to produce different grades of branches and different sizes of side chains.

Expression and protein structure–activity analysis in the plant kingdom indicated that SBE plays a critical role in affecting the fine structure of amylopectin (Satoh et al., 2003). In another study of developing lotus seeds, a similar conclusion was drawn. In this study, we learned that NnSBEI belong to the SBE B family and tend to transfer longer chains, while NnSBEIII belonged to the SBE A family and prioritized short chain. Higher catalytic activity and expression level of NnSBEI indicated that the long chain was preferred in the transfer during the starch synthesis of lotus seed. Zheng, Zheng & Zeng’s (2004) research about the molecular structure of lotus seed found that the branching degree of amylopectin was short, and the glucose residues of side chains were longer, more than 30. This is consistent with the level of genetic research on NnSBEs. Therefore, transcription and activity of SBEs in lotus have a great influence in the fine structure of amylopectin. Furthermore, the edible quality of lotus seed was affected, making it easy for starch to retrogradate.

Conclusion

This study undertook the preliminary study of NnSBE genes in lotus. Two isoforms which encoding starch branching enzyme, were isolated and characterized from lotus. Genetic diversity was analyzed by SNPs of two NnSBEs, and revealed the genetic variation levels among varieties. Difference of expression patterns and the affinity about two isozymes were described, as well as the enzyme activity of SBE in development and different tissues, provided necessary information for understanding of the processes involved in starch synthesis from the level of gene and protein. This study revealed the selection of NnSBE genes during the cultivation process of lotus, and the effect of NnSBE genes on the fine structure of starch in lotus seed. Although the relationships between transcription level, enzyme activity and starch accumulation are complex, our study provides us as much functional information as possible.

Supplemental Information

Supplemental Information 1 The names and SNPs of 45 individual lotus of four clusters

Click here for additional data file.

Additional Information and Declarations

Competing Interests

Author Contributions

Data Availability

The authors declare there are no competing interests.

Fenglin Zhu conceived and designed the experiments, performed the experiments, analyzed the data, contributed reagents/materials/analysis tools, prepared figures and/or tables, authored or reviewed drafts of the paper, approved the final draft.

Han Sun performed the experiments, approved the final draft.

Ying Diao authored or reviewed drafts of the paper, approved the final draft.

Xingwen Zheng and Keqiang Xie contributed reagents/materials/analysis tools, approved the final draft.

Zhongli Hu conceived and designed the experiments, approved the final draft.

The following information was supplied regarding data availability:

The SNP data is available in the Supplemental File.

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
