# Peer review of "Genetic diversity, functional properties and expression analysis of NnSBE genes involved in starch synthesis of lotus (Nelumbo nucifera Gaertn.)"

_PeerJ, doi:10.7717/peerj.7750_

## Round 0.1 · original submission · Major Revisions

Dear Zhongli,

Thank you for your submission to PeerJ.

It is my opinion as the Academic Editor for your article - Genetic diversity, functional properties and expression analysis of NnSBE genes involved in starch synthesis of lotus (Nelumbo nucifera Gaertn.) - that it requires a number of Major Revisions.

My suggested changes and reviewer comments are shown below and on your article 'Overview' screen.

If you address these changes and resubmit, there's a good chance your article will be accepted (although this isn't guaranteed).

Although not a hard deadline, we expect you to submit your revision within the next 55 days.

With kind regards,
Gilda Eslami
Academic Editor, PeerJ

Reviewer 1 ·

Basic reporting

Good in all indicated aspects

Experimental design

Acceptable

Validity of the findings

Findings are valid

Additional comments

Scientific design and writing is within the standard frame

Reviewer 2 ·

Basic reporting

no comment

Experimental design

no comment

Validity of the findings

no comment

Additional comments

The authors identified two isoforms of NnSBEs, analyzed their genetic polymorphisms and variation, determined their spatial and temporal expression, and investigated their effects on amylopectin structure. This study can provide some fundamental information for regulating the starch content and structure in lotus. I recommend accepting it for publication after minor revision.
1. It is better to concise the Section of Introduction.
2. In the Section of Conclusion, it is better to give your main conclusion and the significance of the study, and avoid citing references and discussing them.

Reviewer 3 ·

Basic reporting

The manuscript reads well, contains useful information mainly for those interested in starch accumulation in scared lotus.

Experimental design

no comment

Validity of the findings

The connections between the results and conclusion should be enhanced.

Additional comments

Zhu et al. identified two SBE family genes, NnSBEI and NnSBEIII in sacred lotus (Nelumbo nucifera), analyzed the phylogenetic tree, enzyme activity and the expression profiles. The experiments were well designed. The study will provide some information for the research of starch in sacred lotus. However, the connections between the results and conclusion should be enhanced.

1. In the “Abstract”, the authors concluded that “NnSBE genes received artificial selection during the process of cultivation and domestication in lotus seeds.” However, in the present study, only 6 SNPs and 45 lotus individuals were performed. More results and statistical analysis should be provided to validate the artificial selection of NnSBE in sacred lotus (Nelumbo nucifera), such as the SNPs from upstream and downstream of the NnSBE and more lotus individuals or whole genome resequencing data and so on.
2. In the study, the authors identified the two isoforms, NnSBEI and NnSBEIII in sacred lotus to provide some information for the breeding lotus with high starch content. However, the expression profiles of NnSBE showed that the highest transcript levels of NnSBEI and NnSBEIII were observed in leaves (Line 220). As we know, rhizome and seeds of scared lotus play the key role in a daily diet (Line 37~38), how these two genes (NnSBEI and NnSBEIII) will be used for the edible starch content breeding of sacred lotus?
3. In “Results” section, Line 182 “two SBE genes were obtained using RT-PCR and RACE technique”, but there is no RACE used in the “Material and methods”. And the correct amplification of cDNA fragments by electrophoresis should be provided.
4. The authors descripted that “SBEI and SBEIII subunits showed homology in their α-amylase catalytic domain, but highly dissimilar sequences in N-terminus and C-terminus” (Line 288~290). However, no relative results shown that difference. The structures of NnSBEI and NnSBEIII, should be further analysis, especially the difference between them.
5. The authors analyzed genetic polymorphism of NnSBE in 45 individuals of lotus using Sanger sequencing. How many clones were sequenced for each sample? Although the author used high fidelity KOD DNA polymerase, enough clones should be ensure the sequences correct. While, whole genome resequencing data for some of the lotus accessions have been reported (Huang et al., 2018). Using the whole genome data will be useful for the genetic variations and the evolution of the two isoforms of NnSBE.
6. The different haplotypes of NnSBE in different types of scared lotus is useful for breeding (Line 213~214). More results and discussion can be included in the manuscript.
7. Figures should be high quality and well labelled, some examples:

Figure 1 The labels “NnSBE1” and “NnSBE3” should be “NnSBEI ” and “NnSBEIII”.
Figure 6b The horizontal ordinate had a unexpected “A”.
Figure 7b The line chart represents “NnSBEI ” or “NnSBEIII”? should be clear. And the legend was not include (b).

8. There are many grammatical errors and typos throughout the manuscript. The manuscript should be more carefully edited by native English speakers or professional language services to fix the language issue. Not only include the following examples:

Line 66~67 “while SBE B family showa higher affinity for amylose.”
Line 77 “To study SBEs further”
Line 86 “Nelumbo nucifera. cv. Taikong lotus 36” should be italics. There are many other similar cases.
Line 151~152 “Cheng et al.’s protocol of measuring peak period of SBE activity during the rice grain filling stage was used in this study” here the “in this study” is easy to misunderstand.
Line 261~263 “finding that rhizome lotus had the lowest genomic diversity and a closer relationship to wild lotus, whereas the genomes of lotus seed and flower were admixed”, here the “rhizome lotus” or “lotus seed and flower” is unambiguous? What they are mean?
Line 274 “The results of q-PCR showed that the two NnSBEs were expressed throughout plant tissues. ”, here the “q-PCR” should be “qRT-PCR”.

Reviewer 4 ·

Basic reporting

Lotus is now a very popular vegetable in Southeast Asia. Starch is one of the major nutrients in lotus seed and rhizome, the two major organs that are consumed as vegetables. To know the mechanism of stach biosynthesis in these tissue might be very helpful in lotus breeding and cultivation. This manuscript focusing on one of the critical enzymes SBE that contribute greatly for the biosynthesis of amylopectin. Valuable information could be obtained. However, before it could be accepted, the authors may have to concern about the following questions and comments.
1, The writing is not good in English. It should be polished by some native speakers.
2, Line 46: in the synthesis of amylose and amylopectin.
3, Line 79: “using genes and protein sequences of the two NnSBEs”. In this sentence, it might be better to use DNA rather than genes.
4, The Latin name of lotus “Nelumbo nucifera” should be italic. Furthermore, when the “SEB” was used to stand for gene, it should also be italic
5, There should be a space between the digital number and the unit, eg. 5 μl.
6, In table 1, it might be better to mark the forward and reverse primers. Table 2, amino acid (Aa) should be number of amino acids.
7, Figure legends for figures 2-7 should be revised, to provide more suitable and detail information.

Experimental design

It is ok.

Validity of the findings

Acceptable with some new information.

Additional comments

As stated above.

---

## Round 0.2 · Minor Revisions

Dear author,
,
Thank you very for the revision and re-submission.

Please revise some points in your manuscript as below and then re-submitted again:

1) Correction of the gene name as italics

2) Addition of some more information to the legends for figures 5, 6 and 7, eg. how many repeats, and what is the p-value?


Sincerely yours .

Reviewer 2 ·

Basic reporting

no comment

Experimental design

no comment

Validity of the findings

no comment

Additional comments

The authors have revised their manuscript according to my comments. I recommend accepting it.

Reviewer 3 ·

Basic reporting

no comment

Experimental design

no comment

Validity of the findings

no comment

Additional comments

The problem have been well revised.

Reviewer 4 ·

Basic reporting

The authors have answerred nearly all my concerns, but the following two aspects:
1) They still did not spell the gene name in an italic way (eg. SEB A and SEB B, when are used to refer to gene, they should be italic).

2) the legends for figure 5, 6 and 7 should contain some information about statistics. eg. how many repeats, and what is the p-value?

Experimental design

Good

Validity of the findings

Valuable

Additional comments

See the basic report

---

## Round 0.3 · Minor Revisions

Dear Dr. Hu,

It is our pleasure to inform you that your manuscript titled "Genetic diversity, functional properties and expression analysis of NnSBE genes involved in starch synthesis of lotus (Nelumbo nucifera Gaertn.) " is almost ready for Acceptance.

"This manuscript relies on existing public data and extends minimal analysis to describe the affinities of the starch-branching enzymes of lotus. In some respects this may be important to differentiate among lotus types. Extending the annotation would help if a standard annotation method would be established; such as with the use of GO: annotation. As presented, characterization was done at tissue, biological, and molecular levels; all of these can be added to a gene annotation since not only were the lotus sequences, but other repository sequences used. This would provide great value to the readership community.

Within the text a couple of haplotypes are mentioned, but there is no real provision of its state among the other 45 individuals; perhaps a visual display of the haplotype in addition to the dendrogram would be helpful.

Journal manuscripts are often scanned by text-mining software that locates and extracts core data elements, like gene function. Adding standard ontology terms, such as the Gene Ontology (GO, geneontology.org) or others from the OBO foundry (obofoundry.org) can enhance the recognition of your contribution and description. This will also make human curation of literature easier and more accurate. None of this was visible. I would set the manuscript at the ‘minor revision’ level until the visualization and annotation aspects can be addressed.

There are also some minor language issues; which should be easy items to address. Thank you for your contribution. "

Reviewer 3 ·

Basic reporting

No comments

Experimental design

No comments

Validity of the findings

No comments

Additional comments

The authors have revised all my concerns.

Reviewer 4 ·

Basic reporting

It is now acceptable for me.

Experimental design

Good

Validity of the findings

ok

---

## Round 0.4 · accepted · Accept

Dear Dr. Hu,

I am pleased to inform you that your manuscript titled "Genetic diversity, functional properties and expression analysis of NnSBE genes involved in starch synthesis of lotus (Nelumbo nucifera Gaertn.)" has been accepted for publishing in PeerJ.

My best regards,
Dr. Gilda Eslami

Reviewer 4 ·

Basic reporting

It is now ok for me.

Experimental design

ok

Validity of the findings

good

Additional comments

All my concerns have been addressed.